# Downscaling Industrial-Scale Syngas Fermentation to Simulate Frequent and Irregular Dissolved Gas Concentration Shocks

**DOI:** 10.3390/bioengineering10050518

**Published:** 2023-04-25

**Authors:** Lars Puiman, Eduardo Almeida Benalcázar, Cristian Picioreanu, Henk J. Noorman, Cees Haringa

**Affiliations:** 1Department of Biotechnology, Faculty of Applied Sciences, Delft University of Technology, Van der Maasweg 9, 2629 Delft, The Netherlands; 2Biological and Environmental Science and Engineering, Water Desalination and Reuse Center, King Abdullah University of Science and Technology, Thuwal 23955-6900, Saudi Arabia; 3Royal DSM, Alexander Fleminglaan 1, 2613 Delft, The Netherlands

**Keywords:** syngas fermentation, scale-up, scale-down, Euler-Lagrange, CFD, industrial, lifeline, gas-lift, stirred tank, bubble column, bioreactor

## Abstract

In large-scale syngas fermentation, strong gradients in dissolved gas (CO, H_2_) concentrations are very likely to occur due to locally varying mass transfer and convection rates. Using Euler-Lagrangian CFD simulations, we analyzed these gradients in an industrial-scale external-loop gas-lift reactor (EL-GLR) for a wide range of biomass concentrations, considering CO inhibition for both CO and H_2_ uptake. Lifeline analyses showed that micro-organisms are likely to experience frequent (5 to 30 s) oscillations in dissolved gas concentrations with one order of magnitude. From the lifeline analyses, we developed a conceptual scale-down simulator (stirred-tank reactor with varying stirrer speed) to replicate industrial-scale environmental fluctuations at bench scale. The configuration of the scale-down simulator can be adjusted to match a broad range of environmental fluctuations. Our results suggest a preference for industrial operation at high biomass concentrations, as this would strongly reduce inhibitory effects, provide operational flexibility and enhance the product yield. The peaks in dissolved gas concentration were hypothesized to increase the syngas-to-ethanol yield due to the fast uptake mechanisms in *C. autoethanogenum*. The proposed scale-down simulator can be used to validate such results and to obtain data for parametrizing lumped kinetic metabolic models that describe such short-term responses.

## 1. Introduction

Syngas fermentation is nowadays an established process for the conversion of waste gases into chemicals [1,2]. The company LanzaTech successfully commercializes the fermentation of synthesis gas (containing CO, H_2_ and CO_2_) into ethanol, and is currently exploring other products, such as acetone and isopropanol [3]. Although mass transfer limitations have often been accounted as a limiting factor for scale-up of syngas fermentation, such limitations could highly relieved greatly by making products which are bubble coalescence-suppressing, such as ethanol [4].

High product specificity towards ethanol (>90%) is required for successful commercialization [1]. In a process called solventogenesis, *Clostridium autoethanogenum*, the workhorse of industrial-scale syngas fermentation, produces ethanol from syngas (e.g., using Reaction (1) and (2)), while during acetogenesis, syngas is converted into acetate [5,6]. Solventogenesis can be triggered by low extracellular pH [7], by high extracellular concentrations of acetate [8], or by H_2_ supplementation [9].
(1)$$6\mathrm{CO}+3H_{2}O\to C_{2}H_{6}O+4{\mathrm{CO}}_{2}$$
(2)$$6H_{2}+2{\mathrm{CO}}_{2}\to C_{2}H_{6}O+4H_{2}O$$

Since industrial-scale reactors are of considerable size (e.g., 5 m diameter by 25 m height, or ~500 m^3^, is not exceptional), the occurrence of spatial gradients is more of a rule than an exception. Usually substrate gradients occur when the characteristic time of reaction *τ_rxn_* is significantly lower than the characteristic time of transport, which is related to mixing for substrates in the liquid phase via the circulation time (*t_c_*), and to mass transfer for gaseous-phase substrates (*τ_MT_*) [10,11]. For large-scale syngas fermentation, *τ_rxn_* is expected to be much lower (~0.3 s; [12]) than *τ_MT_* (around 10–20 s; [4]) and *t_c_* (~40 s; Appendix A). Furthermore, both hydrostatic pressure differences (3.5 bar at the bottom vs. 1 bar near the headspace) and gas mole fraction differences (e.g., *y_CO_* decreases from 50% to 5% from bottom to top due to consumption) cause a gradient of around factor 35 in saturation concentration, while the volumetric mass transfer coefficient *k_L_a* might vary locally due to turbulent fluctuations, differences in bubble size and gas hold-up [4]. We hypothesize that all of this leads to sizeable dissolved CO and H_2_ concentration gradients, which might have implications for the syngas fermentation performance.

The impact of such concentration differences on *C. autoethanogenum* can be studied with Euler-Lagrangian CFD modelling. In this way, environmental changes, for example in substrate concentration, temperature, and shear stress, can be recorded from the perspective of the microbe (the so-called “lifeline”) [13,14,15]. The cells are simulated as Lagrangian flow-followers (particles) and, when they do not interact with the flow or concentration field (one-way coupling), are used for analyzing the environmental fluctuations occurring in the bioreactor [16,17,18]. Such analyses could be used for the development of scale-down simulators [19,20] or to study cell population heterogeneity [21]. Two-way coupling has to be realized when studying the influence of biomass on the flow or concentration fields [22,23]. This method requires the use of a structured metabolic-kinetic model that could be coupled with the CFD model in a computationally viable fashion [15]. Although very detailed genome-scale metabolic models and kinetic ensemble models are currently available for *C. autoethanogenum* and other acetogens [9,24,25], two-way coupling of these models is currently too computationally intensive for practical application. Development of less-detailed, yet structured, kinetic models by metabolite lumping [26,27,28] is key in studying the influence of *C. autoethanogenum* on the flow and concentration fields more accurately.

Previously was studied how *C. ljungdahlii* would respond to CO gradients in a severely mass transfer limited bubble column reactor [17]. It was hypothesized, based upon Euler-Lagrangian results and in analogy with *Escherichia coli*, that in such a case transcriptional changes were very likely (>84%) to occur, because long-lasting CO limitations would lead to a maintenance-dominated metabolism. Redox-controlled oscillations in biomass-specific uptake and production rates were observed in *C. autoethanogenum* [29], within the timescale of hours, while substrate fluctuations in the order of seconds (~*t_c_*) or minutes are expected at the large-scale. With scale-down simulators (e.g., based on a single vessel, multiple vessels such as stirred tank reactors coupled with plug flow reactors, or microfluidics) the physiological cell response on such short term fluctuations could be studied, so that the large-scale environment as experienced by the cell is reproduced at bench scale [10,11,13,15,30]. The obtained metabolite fluctuations can be used for parametrization of the lumped metabolic models [26].

Several scale-down simulators have been developed and used in recent decades, but the requirements of a scale-down simulator for syngas fermentation have not yet been identified. Since there are many unknowns in the scientific literature regarding kinetics and short-term cell responses, the execution of scale-down experiments that are representative of the large-scale behavior is crucial for advancing the syngas fermentation field. In this study, we propose a scale-down simulator to study industrial-scale syngas fermentation at lab-scale. To exemplify the distinctive applicability of the proposed scale-down simulator, a wide range of industrial biomass concentrations were studied, since this is a major determinant for the dissolved gas concentration. The impact of gas (CO_2_) production on the dissolved gas concentration gradients, and thus possible fluctuations for the microbe, was studied. We used our previous CFD model of an industrial-scale external-loop gas-lift reactor (EL-GLR) and our lab observations to develop and analyses lifelines representative for large-scale syngas fermentation [4,31].

## 2. Methods

### 2.1. Eulerian Concentration Field

#### 2.1.1. Geometry and Flow Field

As a starting point for the simulations, the 3D reactor geometry and computed flow field of the EL-GLR were used, for which the modelling approach was validated on pilot-scale data, as described in [4]. The only change was in the syngas composition, from a 50% CO, 50% N_2_ mixture to a 50% CO, 20% H_2_, 30% CO_2_ (*v*/*v*) mixture [29]. Since the average molar mass of these compositions is similar and the ideal gas law applies, the mass-flow inlet boundary condition of 2.11 kg s^−1^ was kept the same, as well as the absolute headspace pressure of 101 kPa.

Next to the equations for gas and liquid flow, volume fraction and turbulence, the species equations were solved transiently for both phases to obtain the concentration fields, by implementing user-defined functions for both mass transfer and biological reaction (Figure 1) in ANSYS Fluent 2021R1.

#### 2.1.2. Mass Transfer Model

The mass transfer coefficient *k_L_* was computed by taking the maximum value derived from either the Higbie [33] (Equation (3)) or the Lamont-Scott relation [34], the result of the latter corrected for the underestimation of the energy dissipation rate *ε* by the *k*-*ε* model [20] using the pneumatic power input derived from standard correlations [35] and the liquid volume integral of *ε* (Equation (4)). The maximum *k_L_* of species *i* was used to account for both surface layer renewal mechanisms, since high *k_L_* might be obtained in zones with high energy dissipation [36], but transfer in low-turbulent conditions is better approximated by the Higbie relation [37].
(3)$$k_{L,i}=2\sqrt{\frac{D_{L,i}v_{\mathrm{slip}}}{\pi d_{b}}}$$
(4)$$k_{L,i}={0.45D_{L,i}}^{\frac{1}{2}}{\left(\varepsilon_{\mathrm{cor}}/\nu_{L}\right)}^{{}^{\frac{1}{4}}}\;\mathrm{with}\;\varepsilon_{\mathrm{cor}}=f_{\mathrm{cor}}\varepsilon_{\mathrm{local}}\;\mathrm{and}\;f_{\mathrm{cor}}=\frac{P_{in}}{\int_{V_{L}}\varepsilon_{\mathrm{local}}dV_{L}}$$

In order to obtain a realistic mass transfer rate for industrial-scale syngas fermentation, spherical bubbles with constant diameter (3 mm) were assumed, based upon our previous work [4] (Equation (5)). Since coalescence could be suppressed by the presence of surface-active compounds (e.g., ethanol, salts) in syngas-to-ethanol fermentations, small bubbles can be obtained, leading to high mass transfer rates [18,31,38,39].

While our multiphase model accounted for mass loss through interphase mass transfer and gas expansion using the ideal gas law in Fluent’s volume fraction equation [40], we acknowledge that bubble coalescence, break-up, shrinkage by consumption, and pressure-based bubble expansion were not considered by assuming a constant bubble size. Although these factors could have potentially improved the accuracy of the gas phase description, we chose to prioritize realistic gas mass transfer rates and to focus on the biological aspects of our study. Therefore, we opted for a simplified set of equations, similar to those in [17], that were within the scope of our work.

The saturation concentration was calculated considering the local gas phase mole fraction. The pH equilibrium of CO_2_ with carbonate species could increase the gas-to-liquid mass transfer rate in neutral (pH 6–8) and basic conditions (pH > 8); however, this effect can be neglected, as the syngas fermentation process is operated at pH 5. To ensure complete saturation of CO, H_2_, and CO_2_, and achieve steady-state conditions (i.e., statistically stationary) in the average flow and concentration fields, the mass transfer model was run for 1000 s. Although short-time fluctuations occurred, there were no long-term dynamics, as evidenced by the constant rolling average of the global gas hold-up and dissolved gas concentrations.
(5)$$MTR_{i}=k_{L,i}a\Delta c_{L,i}=k_{L,i}6\frac{\varepsilon_{G}}{d_{b}}\left(H_{i}py_{i}-c_{L,i}\right)$$

#### 2.1.3. Biological Reaction Modelling

The biomass-specific CO and H_2_ uptake rates (*q_i_*) were modelled using a recently derived kinetic model [12]. The kinetic models for both CO and H_2_ uptake account for CO inhibition (Equations (6) and (7)), and are based on the models derived by [41] and [42], respectively.
(6)$$q_{CO}=q_{CO}^{\max}\left(\frac{c_{L,CO}}{K_{S,CO}+c_{L,CO}+\frac{c_{L,CO}^{2}}{K_{I}}}\right)$$
(7)$$q_{H_{2}}=q_{H_{2}}^{\max}\left(\frac{c_{L,H_{2}}}{K_{S,H_{2}}+c_{L,H_{2}}}\right)\left(\frac{1}{1+\frac{c_{L,CO}}{K_{I,CO}}}\right)$$

The overall reaction rate *r_i_* is the product of *q_i_* and the biomass concentration *c_X_*, the latter assumed to be spatio-temporally constant, as a continuous process is considered in steady state. The reaction rates were enabled once CO and H_2_ concentrations reached a steady saturation value. Once statistically stationary flow and concentration fields were obtained (after 600 s, the rolling averages of the global dissolved gas concentrations and hold-up remained constant), time-averaged fields were collected over an averaging period of 200 s. Parameters used for computing the Eulerian concentration fields are provided in Table 1.

The influence of microbial CO_2_ production was examined by modelling two extreme cases at 25 g L^−1^ biomass: (1) only CO_2_ consumption by H_2_ catabolism, $q_{{CO}_{2}}=-\frac{1}{3}q_{H_{2}}$, and (2) also including production by CO catabolism: $q_{{CO}_{2}}=\frac{4}{6}q_{CO}-\frac{1}{3}q_{H_{2}}$. The case with the most extensive dissolved gas concentration gradient was subsequently used to study a wide range of industrially relevant conditions, by running simulations with varying biomass concentrations (2, 5, 7.5, 10 and 25 g L^−1^).

The obtained dissolved gas concentrations from the Eulerian simulations for the different biomass concentrations were compared to approximations obtained by a simple ideal-mixing model (*c_L,i_* from Equation (8)), wherein it was assumed that all transferred gas is directly consumed. The ideal-mixing model used the same parameters (Table 1) and uptake kinetics (Equations (6) and (7)), an average pressure (274 kPa), and the volume-average *k_L_a* obtained from the CFD simulations.
(8)$$MTR_{i}={\left(k_{L}a\right)}_{i}\left(H_{i}py_{i}-c_{L,i}\right)=q_{i}c_{X}=r_{i}$$

### 2.2. Lifeline Analysis

Microbial lifelines were obtained for three cases with different biomass concentrations (5, 10 and 25 g L^−1^). Massless Lagrangian particles were injected at the sparger and tracked for a certain number of circulation times $N_{t_{c}}$ and particles $N_{p}$. To account for turbulence effects, Fluent’s discrete random walk model was enabled. While tracking the particles, the current time, position, concentrations and biomass-specific uptake rates were stored in text format every 0.1 s. Data obtained for the first 90 s (approximately one 95% mixing time *t_m_*, Appendix A) were discarded to ensure that the particles were evenly dispersed over the whole reactor volume during the entire analysis.

The lifelines revealed distinct periods of maxima and minima (i.e., peaks and valleys) in both dissolved CO and H_2_ concentrations. Peak and valley periods were defined from the lifelines by comparing the transient concentrations in the lifeline with the Eulerian average dissolved gas concentration: in case the transient concentration was 2 times (for the 5 g L^−1^ case) or 1.5 times (for the other cases) higher or lower than the Eulerian average concentration for at least 1 s, then a peak or valley was assigned, for which the residence time and the average gas concentration were stored. Probability-normalized histograms were calculated subsequently using 100 linearly distributed bins over the whole parameter space (e.g., residence time or average concentration in peak), except for time in the valleys, where the maximum value was capped at 150 s. The circulation time *t_c_* was calculated as the average time between two peaks (Equation (9)):(9)$$t_{c}=N_{p}\frac{t_{\mathrm{lifeline}}}{N_{\mathrm{peaks}}}$$

For the case with 5 g L^−1^ biomass, lifelines were obtained during *t*_lifeline_ = 1000 s (around 23 circulation times) and for *N_p_* = 160,000 Lagrangian trajectories. This resulted in extensive simulation time and data usage, so that the analysis of the full dataset was computationally unwieldy. We determined how many Lagrangian trajectories ($N_{p}$) and circulation times ($N_{t_{c}}$*)* were needed to ensure statistical independence using the Kullback-Leibler divergence (see Appendix A, Appendix A).

### 2.3. Design of a Scale-Down Simulator

The scale-down simulator was designed based on the results of the lifeline analysis (i.e., the probability density functions of concentrations and residence times in peaks and valleys). The goal of this scaled-down system is to reproduce to the best possible degree the residence times and concentrations experienced by microbes in the full-scale system. The starting point was a continuously operated bench-scale stirred tank reactor (CSTR) (see Figure 1 and Appendix A), for which operational conditions were varied to mimic the large-scale environment at several biomass concentrations.

Mass transfer, dilution and consumption rates were modelled for CO, H_2_, CO_2_ and biomass while assuming ideal mixing in the liquid phase (Equation (10)). The evolution of the gas composition in the dispersed phase *y_D,i_* and in the headspace *y_H,i_* were also considered (Equations (11) and (12)) since these could be highly variable during operation at low gas flow rates. The dispersed gas volume *V_G,D_* was determined by approximating the gas hold-up using the method proposed by [45], while volume balancing was done to calculate the headspace gas volume *V_G,H_*.
(10)$$\frac{dc_{L,i}}{dt}=D(c_{L,i,in}-c_{L,i})+{\left(k_{L}a\right)}_{i}\left(H_{i}py_{i}-c_{L,i}\right)+q_{i}c_{x}$$
$$\frac{dy_{D,i}}{dt}=\frac{F_{G,in}}{V_{G,D}}y_{i,in}-\frac{F_{G,out}}{V_{G,D}}y_{D,i}-\frac{V_{L}}{V_{G,D}}{\left(k_{L}a\right)}_{i}\left(c_{L,i}^{sat}-c_{L,i}\right)\frac{RT}{p}\;\mathrm{with}\;$$
(11)$$F_{G,out}=F_{G,in}-\sum_{i=\mathrm{all}\;\mathrm{gases}}\left[{\left(k_{L}a\right)}_{i}\left(c_{L,i}^{sat}-c_{L,i}\right)\right]V_{L}\frac{RT}{p}$$
(12)$$\frac{dy_{H,i}}{dt}=\frac{F_{G,out}}{V_{G,H}}\left(y_{D,i}-y_{H,i}\right)$$

The volumetric mass transfer coefficient *k_L_a* of compound *i* is dependent on the superficial gas velocity and the stirrer speed [46] and was estimated by considering mass transfer enhancement by the broth composition ($f_{broth}=1.5$; [31]), the temperature and the compound-specific diffusion coefficient in water (Equation (13)). The power input was estimated for a Rushton impeller with $P_{0}=N_{Po}n^{3}d_{i}^{5}$ and the geometry used [32,45].
(13)$${\left(k_{L}a\right)}_{i}=f_{broth}\cdot \left[0.026{\left(\frac{P}{V_{L}}\right)}^{0.4}{\left(u_{G,s}\right)}^{0.5}\right]\left(1.022^{\left(T-293.15\right)}\right)\sqrt{\frac{D_{L,i}}{D_{L,O_{2}}}}\;\mathrm{with}\;P=\alpha {\left(\frac{P_{0}^{2}nd_{i}^{3}}{F_{G,in}{}^{0.56}}\right)}^{\beta }$$

The overall gas consumption rate was determined using the local concentrations in the liquid phase via Equations (6) and (7) and the biomass concentration. The biomass growth rate $\mu \cdot c_{X}$ was determined using the model parameters derived in [47] for solventogenic conditions (Equation (14)), while neglecting the maintenance requirements of the biomass. Biomass retention in the system was assumed (e.g., [48]) and varied by adjusting the biomass recycling rate *R_rec_*.
(14)$$\mu =q_{CO}Y_{X/CO}+q_{H_{2}}Y_{X/H_{2}}$$

The modelled bench-scale reactor was operated with a constant dilution rate of 0.021 h^−1^, inlet gas flow of 0.05 vvm, temperature of 37 °C, pressure of 101 kPa and a stirrer speed during start-up of 75 rpm. Initial concentrations of CO, H_2_, CO_2_ and biomass in the liquid (Table 2) were assumed to solve the system with the ode15s function in MATLAB. After the start-up period, the concentration oscillations were repeatedly imposed by varying the stirrer speed. The obtained scale-down lifelines were analyzed using the same routine as for the industrial-scale reactor but considering no threshold factor to discriminate between the peaks and valleys, since these were manually imposed.

## 3. Results

### 3.1. Eulerian Concentration Gradients in the Industrial Reactor

#### 3.1.1. Influence of Gas Production

The results of Eulerian simulations with 25 g L^−1^ biomass in two CO_2_ production cases were compared in terms of the dissolved CO concentration *c_L,CO_* distribution in the reactor (Figure 2a,b). Although the dissolved gas concentrations in both simulations were in the same range (as expected, since the biomass concentration was kept constant), the spatial distribution of *c_L,CO_* within the riser was completely different. In both cases, the highest CO concentrations appeared at the base of the riser, where the mass transfer rates are high due to the hydrostatic pressure and high CO and H_2_ gas fractions. As the gas rises, the pressure and gas fractions decrease, leading to lower mass transfer rates. More mass transfer was observed in the top separator due to the locally increased gas hold-up (Appendix A), leading to increased *c_L,CO_*. In the downcomer, the long biomass residence time and poor gas renewal caused low CO concentrations.

The gas plume is pushed towards the left side by the liquid exiting the downcomer, causing high dissolved gas concentrations at the left side in the case without CO_2_ production, which is due to reduced oscillations in the gas plume. When CO_2_ is considered, the gas, and thus the dissolved syngas, concentrate towards the middle (Figure 2a,b), in a similar manner to the case without gas consumption [4]. Additional gas is generated halfway along the riser by microbial reaction and is transferred back to the gas phase due to CO_2_ oversaturation at decreased hydrostatic pressure. The additional gas in the riser (cf. Appendix A) leads to transport dissolved CO towards the right side of the riser (cf. Figure 2a,b) and homogenizes the dissolved gas distribution (i.e., the variation of *c_L,CO_* and *c_L,H2_*) within the whole reactor volume (Appendix A).

Next to *c_L,CO_*, the evolution of gas hold-up *ε_G_* and consequently *k_L_a* in the EL-GLR are highly affected by gas consumption (Figure 2c,d). As the mass transfer simulation starts without dissolved gas at *t* = 1200 s, there is an initial drop in *ε_G_* due to gas dissolution. The lower *ε_G_* causes a drop in *k_L_a* because of their linear dependence (Equation (5)). After about 400 s, the liquid saturates with dissolved gas and *ε_G_* and *k_L_a* stabilize. When the reaction is switched on at *t* = 2200 s, both *ε_G_* and *k_L_a* suffer a significant drop in cases with high biomass concentration, since high amounts of gas are being consumed (Figure 2c), even when CO_2_ production is included. Similar decreases in *ε_G_* were also visible in the model in [17]. Interestingly, with little biomass in the reactor (2 g L^−1^), the gas conversion decreases significantly (from 0.67 kg s^−1^ at 25 g L^−1^ to 0.16 kg s^−1^) due to inhibiting CO concentrations (see Section 3.1.2 and Appendix A), and increasing *ε_G_* and *k_L_a,* compared to the cases with more biomass.

Although *ε_G_* could be well predicted with empirical relations in cases without gas consumption [4], the 33% decrease in *ε_G_* by microbial gas consumption makes the prediction of *ε_G_*, and thus *k_L_a*, even more challenging in operational EL-GLRs. This observation is especially relevant for gases rich in carbon source or electron donors, like the used syngas, in contrast to air, where the dilution with inert N_2_, and typically near equimolar conversion of O_2_ into CO_2_ results in negligible volume changes due to mass transfer.

The reduced gradients when considering CO_2_ production make the *c_L,CO_* variations less impactful for the micro-organisms. Due to uncertainties regarding the metabolism, e.g., the possibility of simultaneous CO and H_2_ consumption [49], the modelled cases would either under- or overestimate the CO_2_ production rate. Since the case without CO_2_ production appears to generate larger fluctuations and thus complicate the design of the scale-down simulator (and this is, dependent on the syngas composition, the ideal gas fermentation process from an environmental point of view), we chose to further examine this scenario.

#### 3.1.2. Influence of Biomass Concentration

Dissolved gas concentration fields in the large-scale reactor were computed with 2, 5, 7.5, 10, and 25 g L^−1^ biomass, as shown in Appendix A. The variability that the microbes experience in dissolved CO and H_2_ concentrations, as well as the corresponding biomass-specific uptake rates *q_CO_* and *q_H2_*, are displayed in Figure 3.

The mean values for the Eulerian fields follow the same trend as the ideal-mixing model (Equation (8)), indicating that the concentration range is predominantly $c_{L,i}<K_{S,i}$. For *c_X_* below 5 g L^−1^, the high potential mass transfer capacity compared to the reaction rate leads to strong CO inhibition, while in the 5–10 g L^−1^ range, the mass transfer rate is in equilibrium with microbial syngas consumption at lower dissolved gas concentrations. At high *c_X_* (25 g L^−1^), gas uptake is fast, leading to low *c_L,i_* and thus also to low uptake rates. This decrease in *q_i_* is compensated by the greater *c_X_*, causing the volumetric reaction rate and gas conversion to remain similar to cases with less biomass and higher biomass-specific uptake rates (Appendix A). The ideal-mixing model suggests that an optimum *q_i_* could be obtained at a certain biomass concentration, but the exact biomass concentration remains difficult to be determined using the CFD models, considering the wide concentration distribution, the non-linear kinetics, and that iteratively running these models is very time-consuming. As there is less inhibition at higher *c_X_*, there could be a possibility of increasing gas conversion by supplying more gas, providing that coalescence remains suppressed by the broth components [31].

There is a large volumetric spread in the dissolved gas concentrations obtained by the CFD models. The highest quartile of concentrations is often a factor of 10 higher than the concentrations in the second quartile (e.g., at 5 g L^−1^ Q2 of *c_L,CO_* starts at around 10^−2^ mol m^−3^, while Q4 starts at 10^−1^ mol m^−3^). This would imply that the micro-organism could experience regular concentration fluctuations of around one order of magnitude. However, due to the non-linear nature of the CO and H_2_ uptake kinetics, such fluctuations only lead to minor oscillations in biomass-specific uptake rates. Here, the observed concentration gradients are significantly smaller than those in sugar fermentations with similar *τ_rxn_* [16], due to the continuous gaseous substrate supply. However, the spread in the concentration fields may cause an overestimation of uptake rates by the ideal mixing models.

Overall, the ideal mixing model was able to describe the concentration range reasonably well, especially in the limitation regime, and could still be used for quick estimations of dissolved gas concentrations at varying conditions (e.g., increased mass transfer, pressure, or with different kinetics). Then, the spatio-temporal variations *c_L,i_*, which can only be obtained by CFD modelling, are to be estimated as ± half-an-order of magnitude around the derived concentrations from the ideal mixing model.

### 3.2. Lifeline Analysis

From the lifelines obtained in cases with 5, 10 and 25 g L^−1^ biomass, it appears that the micro-organisms could experience frequent fluctuations in solute concentrations (in 5 to 30 s), as visible from Appendix A. To quantify the microbial experience, the residence times in the peaks and valleys of substrate concentrations were determined, as well as the average dissolved CO and H_2_ concentration during a peak or valley. From the resulting probability density functions, we determined the joint probability of observing a specific residence time and concentration in a peak or valley (Figure 4).

The dynamic behavior of the EL-GLR causes a large spread in the observed concentrations and residence times. This makes it impossible to standardize a concentration profile of a lifeline. The differences in concentration between the peaks and valleys are around a factor of 5, but within these there are significant deviations (up to 50%) around their specific mean values.

Although the average gas concentrations during the oscillations are very different for the three biomass concentrations, the microbial residence time distributions are quite similar. This is caused by the similar hydrodynamic behavior in the three cases, resulting from similar superficial gas flow velocity and gas conversion rates, while neglecting the influence of biomass concentration on fluid properties (density and viscosity). Interestingly, in the cases with 5 and 10 g L^−1^ the dips in concentration lasted sometimes for a very long time (>100 s). This could be due to a recirculation pattern in the downcomer. Since, in the 25 g L^−1^ case, the concentration difference is small between peaks and valleys, and some gas pockets with relative high CO concentration still exist, such moments were not observed.

During the CO peaks at 5 g L^−1^, cells can spend short moments (between 5 and 15 s) at inhibitory concentrations (since *K_I_* = 0.25 mol^2^ m^−6^). In this case *c_L,CO_* still remains at around the values needed for an optimum *q_CO_*, so that strong inhibition would not be expected, since *t_peak_* < *t_c_* ≈ 40 s, based upon the used kinetic model. The precise microbial response to such short moments of potential inhibition is unclear.

Similar results were obtained for H_2_ (Appendix A). The residence times in peaks and valleys were in the same ranges as for CO, with high concentration fluctuations of about a factor of 5 noticed. Because H_2_ uptake was inhibited at a relatively low *c_L,CO_* (*K_I,CO_* = 0.025 mol m^−3^), this resulted in high levels of CO inhibition during peaks. When interested in H_2_ (and thus CO_2_) conversion, CO levels should be kept well below the inhibitory values, which could be achieved by adjusting the inlet gas composition (e.g., by green hydrogen supplementation) and/or by increasing the biomass concentration. Due to the strong fluctuations in *c_L,CO_* and the inhibiting effect of CO on H_2_ uptake, *q_H2_* was significantly influenced, highlighting the need to study the mutual effect of CO and H_2_ fluctuations on *q_H2_*.

The ratio between the average dissolved concentrations of CO and H_2_ (*c_L,CO_*/*c_L,H2_*) increases with an increased biomass concentration: *c_L,CO_*/*c_L,H2_* ≈ 2 at 5 g L^−1^, 3 at 10 g L^−1^ and 6 at 25 g L^−1^. This is caused by the faster decrease of *q_CO_* compared to *q_H2_* with biomass concentration (cf. Figure 3c,d), due to decreased CO inhibition. To inspect the level of inhibition by CO in the determined ranges for peaks and valleys, the biomass-specific CO and H_2_ uptake rates were calculated for each case using its respective *c_L,CO_*/*c_L,H2_* ratio (Figure 5).

From determining the specific gas uptake rates, it became clear that the reactor should be operated in the limitation regime, when increasing *c_L,i_* would result in a greater *q_i_* (e.g., at 25 g L^−1^), while inhibitory concentrations are avoided. At low biomass concentration (5 g L^−1^), CO inhibition is already problematic, leading to decreased H_2_ uptake rates in the peaks. With 10 g L^−1^, a significant increase in *q*_CO_ is observed when transitioning from a valley to a peak (from 0.3 to 0.8 mol mol_x_^−1^ h^−1^), but small increases in *c_L,CO_* during the peaks could worsen overall performance, since *q_CO_* is close to optimum. From the oscillatory dataset in [29], it was derived that fluctuations in *q_CO_* and *q_H2_* in the timescale of hours lead to large increases in *q_EtOH,_* and thus the ethanol yield [12]. Scale-down experiments with imposed concentration fluctuations could be informative as to whether this observation also holds for the circulation timescale.

Too low dissolved gas concentrations would cause a thermodynamically infeasible catabolism and thus no syngas uptake at all. Such concentrations were estimated to be around 4 × 10^−4^ and 3 × 10^−3^ mol m^−3^ for CO and H_2_, respectively, assuming independent consumption of CO and H_2_/CO_2_ for solventogenesis [47]. Since such low *c_L,CO_* was not obtained in our analysis, we do not expect such problems for CO consumption. However, for H_2_, values below the thermodynamic limit were attained in the valleys for 10 and 25 g biomass L^−1^, so that a coupling with CO consumption is potentially required to supply enough electrons for H_2_ uptake. As this may come at the expense of the product yield, further scale-down studies are required to determine how *C. autoethanogenum* may react to such short-term fluctuations in H_2_ concentration.

It has been estimated that a starvation regime could occur when *c_L,CO_* < 3 × 10^−3^ mol m^−3^ [17]. Since, in this regime, a major portion of the energy might be spend on maintenance catabolism, lower growth rates can be expected, leading to higher product yield. In the configuration they studied, such CO shortages were highly likely to occur, causing a probable shift towards a starvation regime. In our simulations, this situation may only occur in the valleys when operating with high biomass concentrations. Due to higher *c_L,CO_* in our other cases, reaching the maintenance catabolism was very unlikely to occur, since *k_L,CO_a* of the EL-GLR was a factor 5 higher than in the BCR studied in [17] (with *k_L_a* ~ 0.01 s^−1^ and *c_X_* = 10 g L^−1^).

Our results suggest that even higher biomass concentrations may be advantageous, considering the current operation in the limitation regime and that high mass transfer could be obtained due to bubble coalescence suppression in the fermentation broths. Operation at very low *c_L,CO_* would enable operational flexibility and a high product yield, without sacrificing gas conversion. Caution is needed to prevent the dissolved gas concentrations becoming so low that the reaction becomes thermodynamically unfeasible, or that the high biomass concentrations hampers mass transfer and mixing by increasing broth viscosity [50].

In the LanzaTech process, *k_L,CO_a* could well be around 3–4 times higher [4] than the final one obtained in our model (650 h^−1^ vs. 180 h^−1^). This could be due to mass transfer intensification (e.g., by introduction of perforated plates [51]) or by achieving smaller bubbles (~1 mm). Although bubbles would become more rigid in such a case, mass transfer might still be enhanced by cell monolayers around the bubbles, especially in case of operating at high biomass concentrations [45]. In such high mass transfer cases, substantial CO inhibition might be expected, stressing the need to operate at high (>25 g L^−1^) biomass concentrations.

### 3.3. Development of Scale-Down Simulator

Based upon the analysis of the CFD data, we used numerical simulations to propose a conceptual design of a scale-down simulator, in order to experimentally replicate the dissolved gas concentrations which were estimated to be experienced by micro-organisms in the industrial-scale syngas fermentation. A single-vessel system, with a 2 L working volume CSTR, was chosen as a basis for the scale-down simulator. Since rapid and irregular fluctuations in peaks and valleys should be obtained, we did not consider multi-vessel systems with forced circulation, because rapid consumption of dissolved gas in the tubes connecting the vessels would be detrimental to performance. In addition to the advantage of no substrate depletion in the tubes, clogging or high shear stress by pump action could also be avoided. In a well-mixed stirred-tank, there are little spatial variations in dissolved gas concentration (unlike plug-flow systems) so that the dissolved gas concentrations can be better controlled. Another potential scale-down system would be a two-stage STR, with a perforated plate separating two well-mixed zones [52]. In this way, the dynamic interchange between two concentrations could be reproduced, although the step transitions might be unrealistic for the observed large-scale behavior. A significant disadvantage of using a single-vessel is the lack of population heterogeneity in cellular experience, which would definitely be present in multi-vessel systems, as well as the poor incorporation of dead (or low concentration) zones, such as the downcomer [53,54]. Furthermore, slow mixing at low power input might possibly lead to local concentration gradients [55]. Operation at smaller scales with better mixing (e.g., 200 mL) might, however, lead to practical problems regarding sampling.

To mimic the large-scale successfully, we should make sure that the microbes experience similar peak/valley durations and the same concentration differences as in the large-scale bioreactor. Although we could argue from Figure 4 that most peaks last between 5 and 15 s, and valleys 10 to 30 s, this argument does not account for frequently occurring irregularities. There are peaks and valleys that largely exceed these times, e.g., peaks of 25 s and valleys of 60 s are not exceptional, and the scale-down simulator should replicate such outliers in terms of time and concentration. With the probability distributions of the residence times derived from the CFD lifelines, variations in the stirrer speed were imposed to obtain corresponding peaks and valleys in the scale-down simulator. It was determined that around 2000 oscillations, lasting ~15 h in total, should be applied in the scale-down simulator to make sure that enough variation in peak or valley residence time is imposed (Appendix A). To account for the varying biomass concentrations from the three CFD cases (5, 10 and 25 g L^−1^), the biomass recycling rate *R_rec_* was altered between the different cases to adjust the reaction rate.

With this computational set-up and the iteratively derived operational conditions (Table 3), lifelines were simulated in the conceptual bench-scale reactor for the three different biomass concentrations that roughly correspond to the large-scale lifelines (Figure 6). The pulses in stirring speed are well captured and provide the same peak-valley frequencies as is expected at the large-scale. The concentrations that the microbes would experience are similar as in the large-scale bioreactor within the peaks and valleys. For example, for CO and H_2_ in the 5 g L^−1^ cases, the upper concentrations are always in the same order of magnitude and the experienced valleys are very similar to those in the large-scale reactor (Figure 6a,b). For the cases with high *c_X_*, it is more challenging to represent the deep concentration valleys well (*c_L,i_* << 10^−3^), since the increased *c_X_* generally requires more mass transfer in the valleys. Since the impact of such concentrations on the biomass-specific uptake rates is small (Figure 5), negligible influence was expected.

The rate of increase in dissolved gas concentration during the transition from a valley to a peak in the scale-down simulator is very similar to that in the large-scale bioreactor: instantaneously, the microbes experience concentration increases of up to around 1–2 orders of magnitude in a matter of seconds (1–5 s). The decrease in the slope at the beginning of the peaks in dissolved CO concentrations can equally be identified in some of the peaks of the CFD lifelines. In the large-scale, this rapid increase is due to the micro-organism travelling instantly into a zone with high mass transfer and thus dissolved gas concentrations, while, in the scale-down simulator, the mass transfer increase responds to the step increase in stirring speed.

The transition from the peak to the valley was found to be more problematic to reproduce in the bench-scale reactor for cases with 10 and 25 g L^−1^. Although the dissolved gas concentration decay is more plug flow-like at the large-scale, immediate decreases back to a representative “valley-baseline” were observed in the scale-down simulator. Simulating a ramped decrease in stirring speed could be helpful in obtaining a more realistic decay in concentrations.

A major factor varying between the two scales is the frequency and magnitude of cellular exposure to shear forces. This was quantified with the energy dissipation/circulation function (*EDCF* = $\frac{P_{\mathrm{total}}}{V_{\mathrm{eff}}}\frac{1}{t_{c}}$) [56,57] while approximating $t_{c}$ as $\frac{1}{4}t_{m}$ [55]. In the scale-down simulator, there is highly varying exposure to shear between peaks and valleys when the cells are close to the impellers (*EDCF* varies from 50 kW m^−3^ s^−1^ and 1 × 10^−4^ kW m^−3^ s^−1^, respectively). In the EL-GLR. there is only a high shear region around the gas plume (*EDCF* ~ 0.06 kW m^−3^ s^−1^), without considering bubble burst. The significantly varying *EDCF* between two scales could be impactful when *C. autoethanogenum* is shear-sensitive, but this was not expected, due to its small size (around 3 μm) compared to the Kolmogorov scale (~10 μm) [58,59].

The correspondence of the results of the scale-down simulator with the large-scale reactor was determined by performing a lifeline analysis. In this way, the probability distributions for the residence times and the concentrations in the peaks and valleys could be compared quantitatively (Figure 7 and Appendix A). Generally, a very good correspondence of the residence time distributions was obtained. To some extent this is logical, as CFD results of these are the inputs of the scale-down simulator, although the limitations of a bench-scale reactor do not guarantee sufficiently good correspondence to be feasible; this indicates the feasibility of imposing rapid stirring speed fluctuations in a well-mixed bench-scale system.

Corresponding concentration distributions were more challenging to obtain, since the ranges of the large-scale CO peak concentrations are very large (0.1–0.25 mol m^−3^, Figure 7a). In the scale-down simulator, the CO peak concentration could not become that high (maximum 0.2 mol m^−3^) so that a narrower range was obtained, which was more skewed towards the lower concentrations. The assumptions and parameters used in the mass transfer and kinetic models make it challenging, however, to rely purely on quantitative results for the concentration fluctuations in the scale-down setup. Using more accurate mass transfer and kinetic models would increase the reliability of our quantitative predictions and thus our conceptual scale-down simulator.

Despite all these limitations, we showed that. with a conceptually relatively simple scale-down simulator, the large-scale dissolved gas concentration gradients for a wide range of biomass concentrations could be reproduced at lab-scale. Model-based tuning of the operational conditions (e.g., stirrer speed, gas flow rate, gas composition) of the scale-down set-up on the probability distribution functions of the large-scale reactor, is a possible strategy to maximize correspondence between the two scales and thereby provide a fruitful basis for representative scale-down of syngas fermentation.

### 3.4. Outlook

Further improvement of the scale-down simulator could lead to even better representations of the large-scale behavior. An optimization routine could help in obtaining the best-fit parameters with the CFD-derived data. The ideal scale-down simulator has as few as possible variable parameters and represents the large-scale behavior for a wide range of conditions. The effect of parameters (e.g., stirrer speed during the start-up phase) could be derived using tools such as principal component analysis during the optimization and in helping to decide whether or not to use the parameter in further analyses.

The quantitative results of the CFD study and the proposed scale-down study are strongly dependent on the process conditions (e.g., headspace pressure, gas fraction), as well as on the kinetic model for CO and H_2_ uptake. For example, the current model for CO-uptake significantly underestimates the maximum specific growth rate ($\mu_{\mathrm{model}}^{\max}$ = 0.03 h^−1^) compared to experimental values ($\mu_{\exp}^{\max}$ = 0.12 h^−1^) [60]. Since the used kinetic models for gas uptake are parametrized using insufficient data, the accuracy of our simulations is decreased and is therefore a major drawback of the study. Development of accurate kinetic models is crucial for reliably modelling bioreactors, and we hope that our work motivates further research in this area. The MATLAB scripts describing the conceptual scale-down simulator are openly available and can be used for further development with updated sub-models.se.

Despite the limitations, the proposed set-up and method are still applicable to a wide range of conditions. Even without using CFD, but with an ideal mixing model (Equation (8)), one can estimate the effect of process variables (e.g., increased mass transfer rates) on the average concentration in the large-scale reactor. In case the reactor is operated in the gas limitation regime ($c_{L,i}<K_{S,i}$), spatial concentration differences of a factor of 5 around the mean could be expected based upon our CFD simulations. Since, in the scenarios with varying biomass concentrations the residence time distributions in the peaks and valleys do not greatly differ, neither are such differences expected in other operational cases in the limitation regime. A scale-down simulator could then be tuned based upon the estimated concentration differences and the residence time distributions.

Although the developed scale-down simulator is conceptually easy to understand, practical installation and operation might be challenging. The repeated variations in stirrer speed at high frequencies should be controlled, along with possible ramp phases when increasing or decreasing the stirring speed. Ideally, rapid-sampling and/or online measurement for CO and H_2_ [61,62] should be applied to make sure that peaks and valleys are obtained in the intended manner. As the peaks and valleys are applied in the second-timescale, probe lag should be taken into account when analyzing the experimental data. Furthermore, the influence of broth components on *k_L_a* in real fermentations should be considered for better predictions [31]. If the in-situ concentrations cannot be measured, they could be predicted using a precisely determined *k_L_a* [63].

Our conceptual scale-down simulator makes it possible to simulate a statistically representative lifeline of the EL-GLR within a fraction of the time that it would take for the CFD model to be run and analyzed. Such a lifeline could then be used to study interactions between the extra- and intracellular environment by coupling with a metabolic kinetic model [12]. The obtained response should then be similar to the large-scale response, due to the correspondence of both lifelines. By making variations in such lifelines, the peak and valley residence time and concentration distributions can be obtained, which could lead to a desired large-scale response (e.g., high ethanol specificity).

Ramp and feast-famine studies in the scale-down set-up could be used to parametrize kinetic models that describe the short-term response of *C. autoethanogenum* [12,28,64], by rapid sampling of metabolite and enzyme concentrations. Ramp studies would be helpful in determining whether the instantaneous electron supply in the peaks would indeed lead to increased ethanol production, as was expected from long-term oscillations [29]. If the gas uptake rates are product-independent, then such scale-down simulators could be used for engineered strains to produce higher-value products [3], proteins [65], or for coupled reactions with other micro-organisms, such as chain-elongators [66] or PHA production [67]. With the scale-down simulator, the microbe could be adapted to large-scale conditions, so that fewer scale-up problems might be expected [14].

The analyses of the industrial syngas fermentation process in this and our previous study [4] are all model-based and only slightly tuned, based upon the scarcely available literature data of the full-scale LanzaTech operation [1,2,68]. To advance the syngas fermentation process, for model validation and the execution of highly representative scale-down simulators, the publication of real industrial data would be required, such as large-scale circulation times, operational *k_L_a* values, a range of dissolved gas concentrations and their gradients. All of this could, for example, enable the utilization of a broader range of gas compositions, the development of processes towards higher-value products and intensified fermentation equipment.

In our analyses we showed that high biomass concentrations (e.g., *c_X_* > 10 g L^−1^) might be advantageous for both product yield and gas conversion. Since the highest reported biomass concentration in syngas fermentation reactors is around 9 g L^−1^ [48], experiments should target the influence of increased biomass concentrations and its viability on gas uptake, broth viscosity and mass transfer. The precise operating interval in terms of dissolved gas (CO and H_2_) concentrations should be retrieved experimentally, so that the thermodynamically infeasible range can be avoided while operating in the maintenance-dominated regime, which would enable high product-to-substrate yields. Our results show that, with the currently used syngas composition (20% H_2_) and high biomass concentration (25 g L^−1^), H_2_ catabolism may be thermodynamically infeasible, although co-consumption with CO might occur [9]. Thus, in a future with intermittent green hydrogen supply from renewable resources [69], supplementation of hydrogen might be a good option to valorize excess electricity and increase the CO-to-product yield.

## 4. Conclusions

The effect of biomass concentration and dissolved gas concentration fluctuations in large-scale syngas fermentation was studied with Euler-Lagrangian simulations. Based upon these numerical simulations, we recommend industrial operation at relatively high biomass concentrations, as this would reduce the effects of CO inhibition, could increase the product yield and would provide high operational flexibility. Simulations indicate that, in large-scale syngas fermentation, *C. autoethanogenum* will experience frequent oscillations (peaks and valleys) in a dissolved gas (CO, H_2_) concentration of about one order of magnitude, in a timescale of seconds (5 to 30 s). Such concentration fluctuations may occur irrespective of the biomass concentration and were hypothesized to favor the ethanol yield.

The large-scale concentration fluctuations should be simulated during small-scale experiments to study how *C. autoethanogenum* adapts to industrial-scale conditions. We proposed a single-vessel scale-down simulator that theoretically replicates the fluctuations in dissolved gas concentrations by varying the stirrer speed based on the large-scale oscillations. Numerical analysis shows that the durations of the oscillations could be well replicated, but the settings might be adjusted to achieve higher similarities for the variations in concentration. The obtained lifelines in the proposed scale-down simulator well represent the large-scale reactor for a wide range of biomass concentrations and operational conditions.

## Figures and Tables

**Figure 1 bioengineering-10-00518-f001:**
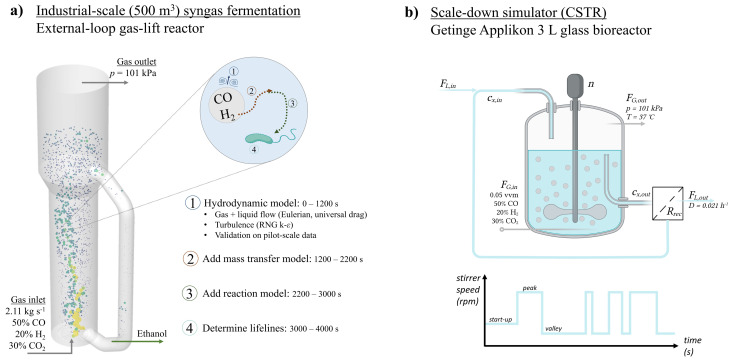
Conceptual representation of the modelling procedure for both (**a**) the industrial-scale reactor and (**b**) the scale-down simulator. The 3D geometry, based on publicly available pictures of the Shougang-LanzaTech plant, the hydrodynamic model CFD of the EL-GLR and its validation on pilot-scale data, are extensively described in our previous work [4]. The scale-down simulator is based upon a 3 L bioreactor (2 L liquid volume), with varying durations of high and low stirrer speed after the start-up phase. See Appendix A for details of the geometry of this reactor [32]. Created with BioRender.com (accessed on 30 August 2022).

**Figure 2 bioengineering-10-00518-f002:**
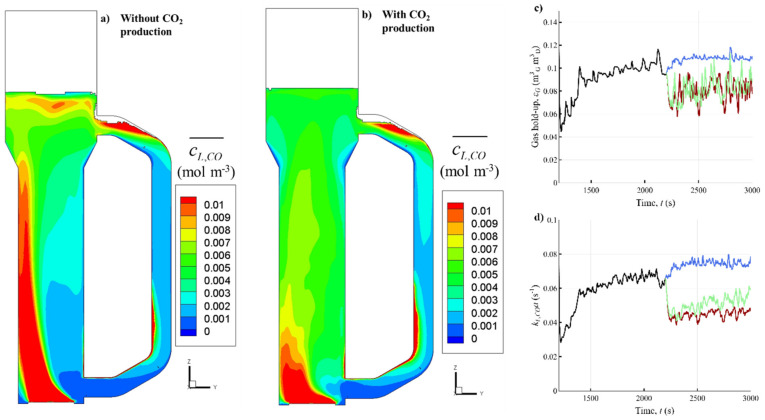
Effect of CO_2_ production on mass transfer and dissolved gas distribution. Time-averaged (200 s) dissolved CO concentrations *c_L,CO_* in the zy-plane (*x* = 0) of the EL-GLR for 25 g L^−1^ biomass (**a**) without CO_2_ production and (**b**) by considering CO_2_ production. Time-dependence of the dispersion volume-averaged (**c**) *ε_G_* and (**d**) *k_L,CO_a* during the CFD-simulations. Until 2200 s, only gas-liquid mass transfer was included (black line). At 2200 s, gas consumption was switched on in the model in three cases: 2 g L^−1^ biomass without CO_2_ production (blue line), 25 g L^−1^ biomass without CO_2_ production (red line), 25 g L^−1^ biomass including CO_2_ production (green line).

**Figure 3 bioengineering-10-00518-f003:**
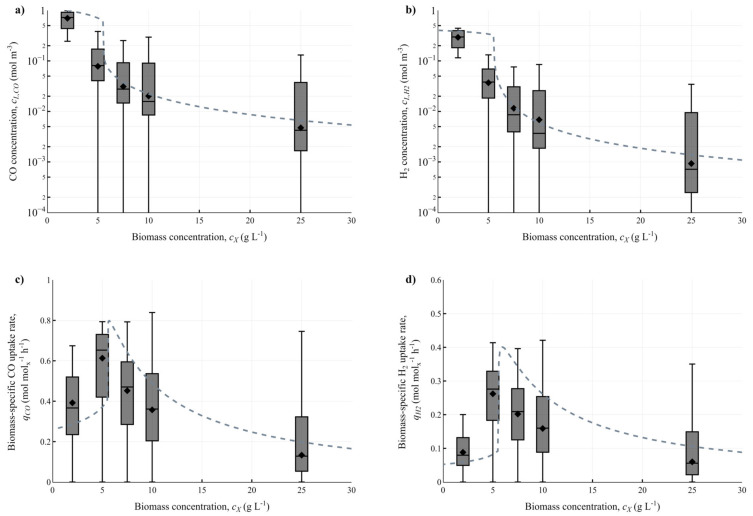
Spatial variations in (**a**,**b**) dissolved CO and H_2_ concentrations and (**c**,**d**) biomass-specific uptake rates, represented in boxplots, for varying biomass concentrations. The boxplots were obtained from the 200 s time-averaged results of the CFD simulations and depict the spread around the mean values, with each quartile representing 25% of the reactor volume, while the diamond symbol represents the volume-averaged mean value. The dashed line is the result of a simple ideal-mixing model, using the volume-averaged *k_L_a* from the CFD model (0.050 s^−1^ for CO and 0.074 s^−1^ for H_2_).

**Figure 4 bioengineering-10-00518-f004:**
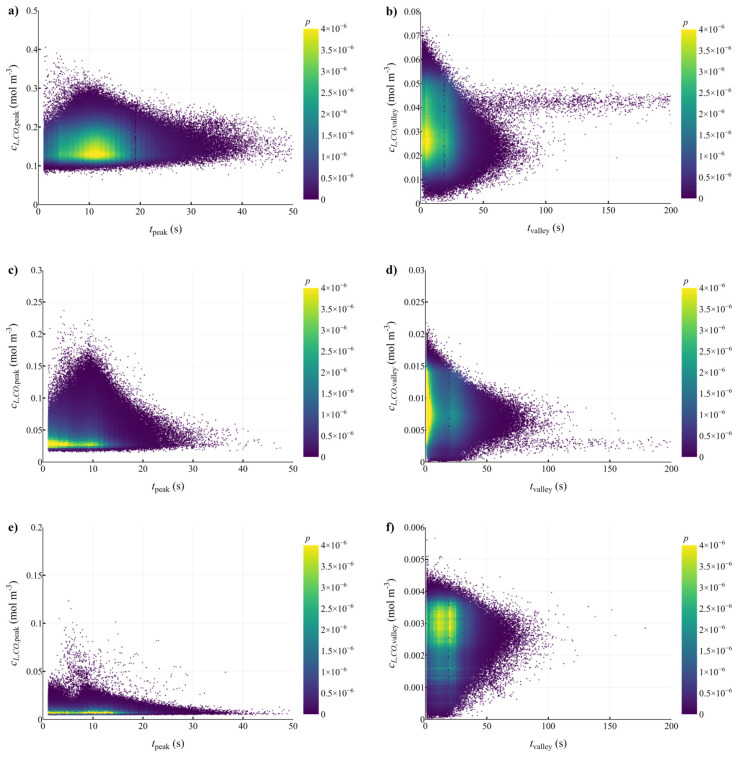
Scatter plots representing the likelihood of a microbe experiencing peaks or valleys along with a certain combination of *c_L,CO_* and duration. Each dot represents a peak or valley with a concentration and residence time, and is colored by the probability of occurrence of that specific combination in the whole set of lifelines. Each row of plots represents data obtained with a specific biomass concentration: (**a**,**b**) 5, (**c**,**d**) 10 and (**e**,**f**) 25 g L^−1^. Peaks are in the left column of plots (**a**,**c**,**e**) and the valleys are at the right (**b**,**d**,**f**). A similar figure was made for H_2_ (Appendix A).

**Figure 5 bioengineering-10-00518-f005:**
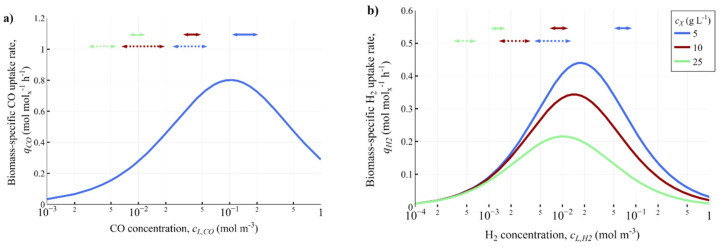
Overall biomass-specific (**a**) CO and (**b**) H_2_ uptake rates computed for biomass concentrations of 5 (blue), 10 (red), and 25 g L^−1^ (green). *c_L,CO_* in the H_2_ uptake kinetics was calculated by using a *c_L,CO_*/*c_L,H2_* of 2, 3 and 6 for each respective case. The full arrows indicate the concentration ranges of the peaks, while the dashed arrows indicate the ranges for the valleys. CO-uptake is independent of the biomass concentration, hence the single line.

**Figure 6 bioengineering-10-00518-f006:**
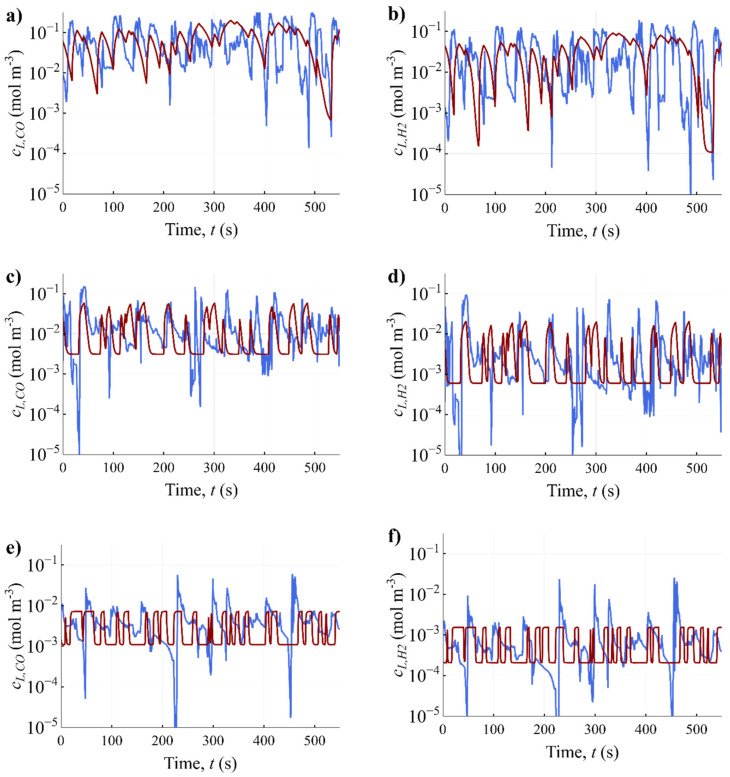
Microbial lifelines obtained from the simulation of the EL-GLR (blue) and the scale-down simulator (red), for CO and H_2_, and the varying biomass concentrations in the large-scale reactor: (**a**,**b**) 5 g L^−1^; (**c**,**d**) 10 g L^−1^; (**e**,**f**) 25 g L^−1^. Random lifelines were chosen from the CFD simulation (blue) and a random timespan of the lifelines in the scale-down simulator (red).

**Figure 7 bioengineering-10-00518-f007:**
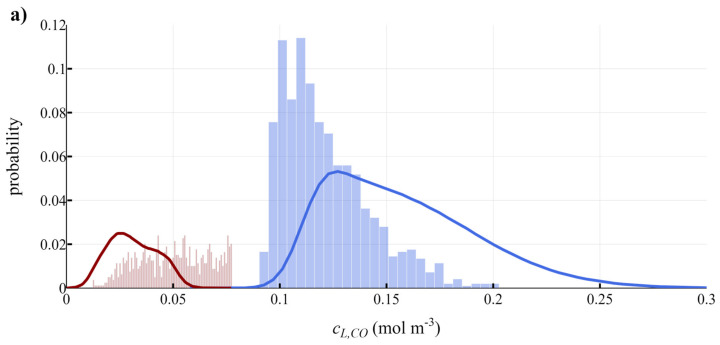
Comparison of the probability density functions obtained by the scale-down simulator (bars) with the CFD results (lines). Probability density functions for (**a**) the concentration of dissolved CO during the peaks (blue) and the valleys (red), as well as the residence time in a (**b**) valley or (**c**) peak. Here, the case of CO with 5 g L^−1^ biomass is provided, other cases (H_2_ 5 g L^−1^, and both compounds for 10 and 25 g L^−1^) are provided in Appendix A. The scale-down simulator was operated with 2000 peaks.

**Table 1 bioengineering-10-00518-t001:** Parameters used for the simulation of the Eulerian concentration field.

Name	Symbol	CO	H_2_	CO_2_	Unit	Source
Inlet gas fraction	*y_i,in_*	0.5	0.2	0.3	mol_i_ mol_G_^−1^	[29]
Henry coefficient	$H_{i}$	2.30 × 10^−7^	1.47 × 10^−8^	1.06 × 10^−5^	kg m^−3^ Pa^−1^	[43]
Diffusion coefficient	$D_{L,i}$	2.71 × 10^−9^	6.01 × 10^−9^	2.56 × 10^−9^	m^2^ s^−1^	[44]
Maximum uptake rate	$q_{i}^{\max}$	1.459	2.565	-	mol mol_X_^−1^ h^−1^	[12]
Half-saturation coefficient	$K_{S,i}$	0.042	0.025	-	mol m^−3^	[12]
Inhibition coefficients	$K_{I}$ $,\;K_{I,CO}$	0.246	0.025	-	mol^2^ m^−6^, mol m^−3^	[12]

**Table 2 bioengineering-10-00518-t002:** Parameters specifically used for the design of the scale-down simulator. Other parameters as in Table 1.

Name	Symbol	CO	H_2_	CO_2_	Biomass	Unit	Source
Inlet gas fraction	$y_{i,in}$	0.5	0.2	0.3	-	mol_i_ mol_G_^−1^	-
Inlet liquid concentration	$c_{L,i,in}$	0	0	0	$c_{L,X,out}R_{rec}$	mol m_L_^−3^	-
Biomass yield	$Y_{X/i}$	0.041	0.0070	-	-	mol_X_ mol_i_^−1^	[47]
Initial liquid concentration	$c_{L,i,0}$	0.1	0.03	7.4	2.03	mol m_L_^−3^	-

**Table 3 bioengineering-10-00518-t003:** Operational conditions of the scale-down simulator to obtain an acceptable fit of the lifelines obtained by the scale-down simulator with CFD-derived lifelines.

	Peak			Valley			Recycle	
$c_{X,\mathrm{EL}-\mathrm{GLR}}$ (g L^−1^)	*n* (rpm)	$k_{L,CO}a$ (h^−1^)	*P/V* (W m^−3^)	*n* (rpm)	$k_{L,CO}a$ (h^−1^)	*P/V* (W m^−3^)	*R_rec_* (-)	$c_{X,\mathrm{SD}}$ (g L^−1^)
5	910	153	23,000	20	1.3	0.11	0.5	0.54
10	900	150	22,000	150	15	70	0.91	1.44
25	500	71	3400	70	5.6	6.1	0.96	3.27

## Data Availability

The processed data from the lifeline analysis (i.e., probability distributions in peaks and valleys for all cases) as well as the MATLAB model of the proposed scale-down simulator are openly available in 4TU.ResearchData at http://doi.org/10.4121/21655781 (accessed on 30 August 2022).

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
