# Peer review of "Downscaling Industrial-Scale Syngas Fermentation to Simulate Frequent and Irregular Dissolved Gas Concentration Shocks"

_bioengineering, 2023, doi:10.3390/bioengineering10050518_

Round 1
Reviewer 1 Report
The manuscript is clear and well written, and addresses an interesting and important topic in the area of industrial biotechnology. However, there are some issues which need to be addressed before the article is suitable for publication.
Generally speaking it is difficult to provide a detailed assessment of some of the modelling approaches used as there is insufficient information provided in the current manuscript. Not all details need to be included, but a reasonable amount is required such that the paper can be assessed as a stand-alone work.
There are some questions about the methodology used by the authors which need to clarified/addressed before the article is suitable for publication.
The authors should acknowledge that a major limitation of their work is the lack of experimental validation. The modelling exercise is useful and important, however it relies on a large amount of information and assumptions which have not been experimentally validated. Obtaining industrial data is very challenging, however the work could be tested on the pilot scale (e.g. via validation of the hydrodynamic model). Similarly, it would be useful to experimentally test the scale-down simulation and compare the effect of the different conditions on the ethanol production.
Line 115: It would be useful if the authors clarified whether or not the headspace pressure was gauge or absolute pressure.
Line 118: The authors should include a diagram of the system to make it easier for the reader to understand what has been done.
Line 124: Do the authors mean Equation 4 and not Equation 2 here?
Line 129: In Equation 4 exactly what V is needs to be clarified – is this the volume of the cell or the entire simulation domain?
Line 130: The authors need to provide some justification for their choice of bubble size, as this seems to be totally arbitrary. Additionally, it is very well known in the field (e.g. see [1-7] among others) that the presence of surface active compounds (like ethanol and others in the fermentation medium) affects both drag experienced by the bubbles and gas-liquid mass transfer. This needs to be at least acknowledged by the authors.
Can the authors please also clarify if the effect of the changing pressure on the bubble volume has been accounted for? Due to the pressure gradient within the reactor the bubble size will vary with height; this needs to be accounted for in the modelling approach as this has a significant effect on the hydrodynamics of the system.
Line 133: In the aqueous phase carbon dioxide exists in equilibrium with carbonate species, and this has a significant effect on mass transfer for this component as the solubility is no longer entirely governed by Henry’s Law. This needs to be noted in the manuscript, and the authors should note that the approach they have used is a simplification which can introduce significant errors.
Line 134: It is not clear what a ‘statistically stationary flow and concentration field’ is. The authors need to make this explicit.
Line 149: It would be useful to know how long it took to achieve ‘steady’ values of the concentration fields using this modelling approach.
Line 207: Information about the geometry of the STR and other key points used to design the scale-down simulator should be kept in the manuscript and not the supplementary material.
Line 272: Figure S3 should also be included in the manuscript if it is important to the discussion – it is very inconvenient for the reader to have to read the supplementary material to find key information related to the discussion when it can be included in the manuscript.
Line 290: It would improve the discussion to include information about the rate of gas consumption (i.e. in kg/s). If the expansion of the gas phase due to the pressure change has not been accounted for this should also be noted as it will have a significant effect on the process.
Figure 3: This may be an issue with the upload, but the quality of the figure is not particularly good, and it is difficult to read. It is also not clear what the probability here refers to – can this please be clarified.
Table 3: it is not clear why the power input for the peak case for 25 g/L has less rpm but a higher power input than for the other cases – can the authors please explain this?
References:
1. Prins, A. and K. van't Riet, Proteins and surface effects in fermentation: foam, antifoam and mass transfer. Trends in Biotechnology, 1987. 5(11): p. 296-301.
2. Ruzicka, M.C., et al., Effect of surfactant on homogeneous regime stability in bubble column. Chemical Engineering Science, 2008. 63(4): p. 951-967.
3. Vasconcelos, J.M.T., et al., Effect of contaminants on mass transfer coefficients in bubble column and airlift contactors. Chemical Engineering Science, 2003. 58(8): p. 1431-1440.
4. Zahradník, J., M. Fialová, and V. Linek, The effect of surface-active additives on bubble coalescence in aqueous media. Chemical Engineering Science, 1999. 54(21): p. 4757-4766.
5. Camarasa, E., et al., Influence of coalescence behaviour of the liquid and of gas sparging on hydrodynamics and bubble characteristics in a bubble column. Chemical Engineering and Processing: Process Intensification, 1999. 38(4-6): p. 329-344.
6. Jamialahmadi, M. and H. Müller-Steinhagen, Effect of alcohol, organic acid and potassium chloride concentration on bubble size, bubble rise velocity and gas hold-up in bubble columns. Chemical Engineering Journal, 1992. 50(1): p. 47-56.
7. McClure, D.D., et al., Towards a CFD model of bubble columns containing significant surfactant levels. Chemical Engineering Science, 2015. 127: p. 189-201.
Reviewer 2 Report
The manuscript describes investigations of biomass concentration and dissolved gas concentration in large-scale syngas fermentation with Euler-Lagrangian simulation. Also, they compare the results from the analytical solution (i.e., the Eulerian approach) with the Lagrangian simulation. Based on the study, the authors explored concentration fluctuations under various conditions. Moreover, the authors proposed a single-vessel CSTR to scale down the system. Actually, all the methods and approaches are not new. Still, it is very worth mentioning that the authors did a crucial exploration with various conditions and down-scaling for the syngas fermatation. Moreover, the reviewer was impressed by the study on the extensive numerical studies with various methods and quantifications. All contents in the manuscript were well organized. However, the contents are so long, which makes the reader tired due to the extensive information. Therefore, the reviewer recommends one or two sections in the results section could be omitted or moved to the supplementary for the sake of a concise manuscript. That so, the reviewer highly recommends the manuscript be published in Bioengineering.
Author Response
Dear reviewer,
It is a pleasure to receive your feedback on the manuscript, we are grateful for the effort you spend on reading and reviewing the manuscript. We agree that the methods and approaches used in our study are not novel, and were applied in past works; we demonstrate their usefulness in a completely different case.
We acknowledge that the results section of the paper is quite lengthy. We had a critical look whether all elements of the results section constitute to the storyline, shortened several sentences and sections, and decided to move the section 3.3 about the statistical analysis of the number of particles and mixing times as well as the corresponding part of the methods section (lines 192-200) to the Supplementary material.
Round 2
Reviewer 1 Report
It would be easier if the authors uploaded a pdf document with their comments as the formatting from the reference management software has made it very difficult to read their reply.
The first set of comments highlighted significant issues with the physics implemented in the CFD model particularly with respect to how the gas phase was modelled. At a minimum the authors need to clearly state that by neglecting key phenomena (i.e. expansion of the gas phase) it is likely that the model used will not correctly capture the behaviour of the system. This is a likely explanation for the observed discrepancy between the predicted and experimental growth rates noted in the paper. Ideally the simulations would be repeated with more realistic/correct physics included to avoid these issues.
Major comments:
Line 137: In equation (4) how is the value of the constant fcor determined?
Line 144: The authors have stated that “Coalescence, break-up, shrinkage by consumption and the pressure-based expansion of bubble were unaccounted for, although the multiphase model accounts for expansion of the gas-phase in the volume fraction equation.” – these statements are contradictory. If the expansion of the gas is accounted for in the volume fraction equation then the bubbles have to expand (or the number of bubbles needs to increase).
Can the authors please identify which equation they mean by the volume fraction equation, and how the expansion is accounted for? This is unclear and important to the work. Looking at Figure S4 it does not appear this has been accounted for. The gas volume is going to change by a factor of approximately two, and this will introduce significant changes in both the momentum and mass transfer terms. Failing to account for this introduces significant error in the simulations, and this needs to be made clear for the reader.
Similarly, the fact that the loss of mass from the gas phase is not accounted for suggests that the modelling approach used may not conserve mass. It would be useful to know the concentrations of the gas components at the outlet, to understand whether or not the changes can be reasonably neglected.
Line 164: Should ‘statically’ should be ‘statistically’ here? The authors also need to make explicit what they mean by ‘statistically stationary’. Do they mean the mean value is constant with time? Have the used an alternative metric to justify this? When they quantify the flow field is this based on some global metric (e.g. the overall hold-up?) or some local metric? This information needs to be made explicit.
Line 285: The authors have not provided information about gas uptake/transfer rates (in kg/s or equivalent units). This information is key in quantifying the mass transfer performance of the system, and comparing their simulations with commercial processes. Such information must be provided.
Minor comments:
- Equations (1) and (2) – the proper chemical formula for ethanol should be used, not the abbreviation.
